# Treatment of Resectable Gallbladder Cancer

**DOI:** 10.3390/cancers14061413

**Published:** 2022-03-10

**Authors:** Eduardo A. Vega, Sebastian Mellado, Omid Salehi, Richard Freeman, Claudius Conrad

**Affiliations:** 1Department of Surgery, St. Elizabeth’s Medical Center, Boston University School of Medicine, Boston, MA 02135, USA; eduardo.vega@steward.org (E.A.V.); omid.salehi@steward.org (O.S.); richard.freeman2@steward.org (R.F.); 2Tufts University School of Medicine, Boston, MA 02111, USA; sebastian.mellado@tufts.edu

**Keywords:** gallbladder cancer, incidental, resectable, oncologic extended resection, treatment

## Abstract

**Simple Summary:**

Gallbladder cancer (GBC) is the most common biliary tract cancer worldwide, with incidental GBC being the most common presentation of resectable gallbladder cancer today. While surgery remains the only curative treatment, important progress has been made in the understanding of molecular pathogenesis and systemic treatment options.

**Abstract:**

Gallbladder cancer (GBC) is the most common biliary tract cancer worldwide and its incidence has significant geographic variation. A unique combination of predisposing factors includes genetic predisposition, geographic distribution, female gender, chronic inflammation, and congenital developmental abnormalities. Today, incidental GBC is the most common presentation of resectable gallbladder cancer, and surgery (minimally invasive or open) remains the only curative treatment available. Encouragingly, there is an important emerging role for systemic treatment for patients who have R1 resection or present with stage III–IV. In this article, we describe the pathogenesis, surgical and systemic treatment, and prognosis.

## 1. Incidence

Although rare in Western countries, gallbladder cancer (GBC) is the most common biliary tract cancer worldwide [1]. Indeed, northern India and Latin American countries such as Mexico, Bolivia, and Chile [2] experience a high incidence of GBC. Gallstones [3], sex, age, obesity [4], parity, [5] chronic gallbladder inflammation, [6] adenomas [7,8], and an anomalous pancreaticobiliary junction [9,10,11,12,13] are the most commonly reported risk factors associated with GBC. Today, most cases of resectable GBC are diagnosed as incidental gallbladder cancer (IGBC), following elective or emergent cholecystectomy [14]. With an increase in elective cholecystectomies, there is a rising number of IGBC cases, where 0.7% to 1.2% [15,16,17] of cholecystectomy specimens contain previously unrecognized gallbladder cancer [1,18,19].

## 2. Gallbladder Cancer Pathogenesis

Chronic biliary inflammation and cholestasis are the basis for gallbladder cancer development [20]. The activation of inflammatory pathways such as the cyclooxygenase-2 (COX-2) pathway and mediators such as nuclear factor-kappa B, cytokines (interleukin 6, IL-6) induce nitric oxide synthase in gallbladder cells. These inflammatory mediators predispose to mutagenesis, impaired DNA repair, DNA methylation, and angiogenesis to promote gallbladder oncogenesis. In addition, bile acid from cholestasis promotes accelerated cholangiocyte growth through the activation of growth factors, which increase cell turnover and result in clonal proliferation via oncogenes (e.g., epidermal growth factor receptor (EGFR), RAS/mitogen-activated protein kinase (MAPK), interleukin (IL-6) and tyrosine kinase receptors such as Met (c-MET)). Moreover, genetic and epigenetic regulators in cholangiocytes or stromal cells lead to evasion of apoptosis, limitless replication potential, neoangiogenesis, invasion, and metastasis [6,21,22].

HER2/neu gene amplification and mutations in the EGFR, KRAS, and PI3K genes [6] represent the most frequent molecular abnormalities in GBC. Of note, these genes are involved in metabolic pathways susceptible to targeted therapy (e.g., Sotorasib for KRAS mutations [23], Erlotinib for EGFR mutations [24], and Transtuzumab or Afatinib for HER2 mutations [25,26]. An effort to more consistently genotype GBC and apply the targeted therapy needs to be further studied to demonstrate clinical applicability.

A comprehensive genomic profiling of 760 GBC patients [27] revealed that the most frequently altered genes were tumor protein 53 (TP53; 61%), cyclin-dependent kinase inhibitors 2A (CDKNA; 28.6%) and 2B (CDKN2B; 18.2%), AT-rich interactive domain-containing protein 1A (ARID1A; 16.4%), SMAD4 (15.8%), ERBB2 (13.9%), KRAS (13.2%) and phosphatidylinositol 3-kinase CA (PIK3CA; 13.4%). From the entire cohort, 14.2% of GBC patients were found to have direct DNA repair gene alteration, and ATM was the most frequently altered gene. Further, the study showed the median tumor mutational burden (TMB) of the 760 patients was 2.6 mut/mb. Of note, tumor mutational burden high (TMB-H) is associated with better response to immunotherapy; however, in this study, TMB-H (define as 19.5 mutations/mb) was present in only 1.2% of the samples.

Moreover, there are well-defined molecular changes that follow the progression of normal gallbladder tissue to invasive carcinoma. Mutations in TP53 and mitochondrial DNA, as well as methylation of tumor suppressor genes and COX-2 overexpression, are implicated in the progression from normal gallbladder to gallstones-induced chronic inflammation. Most p53 mutations include missense mutations that lead to the expression of a non-functional p53 protein. Furthermore, COX-2 overexpression leads to prostaglandin-dependent inflammation and cell growth. Loss of heterozygosity at chromosome loci 3p and 8p is associated with the progression of chronic inflammation to dysplasia; wherein 3p and 8p contain consequential tumor suppressor genes. With progression to carcinoma in situ, mutations in FHIT and CDKN2A as well as further loss of heterozygosity at 9q, 18q, and 22q have been observed. Finally, KRAS mutations promoting progression to invasive carcinoma has been observed in the Japanese population [6].

In a recent study, Lin et al. [28] focused on the two major developmental paths suggested for GBC (1) de novo development of GBC or biliary tract intraepithelial neoplasia (BilIN)-independent path, and (2) BilIN-dependent path similar to the traditional adenoma/dysplasia-carcinoma sequence model when GBC coexists with low-grade biliary tract intraepithelial neoplasia (LG-BilIN) and high-grade biliary tract intraepithelial neoplasia (HG-BilIN). In the later path, the LG-BilIN (adenoma)/HG-BilIN (dysplasia) serves as the precursor of GBC through a stepwise progressive manner. One of the main findings of the study was the identification of a higher proportion of loss of heterozygosity (LOH) and the number of mutational events in the BilIN-independent path compared to the BilIN-dependent path (20% vs. 5% *p* < 0.021 LOH and 150 vs. 36 *p* < 0.05 number of mutations acquired). Therefore, the BilIN-independent path GBC split earlier and evolved more independently from LG-BilIN and HG-BilIN in the cancerous niche that harbored extensive LOH and mutations.

## 3. Prognostic Factors for IGBC at the Index Cholecystectomy

The high rate of IGBC indicates that GBC is either difficult to detect before cholecystectomy or frequently not considered as a differential diagnosis. Even after appropriate imaging and macroscopic examination of the specimen in the operating room, GBC frequently goes undetected [29]. For Western GBC, detecting small lesions is particularly difficult due to the high proportion of flat tumors that are embedded in chronically inflamed gallbladder walls (Figure 1A). According to de Aretxabala et al., more than half of the 95 patients in their sample had flat lesions that went undetected upon gross examination by an experienced pathologist [30]. Hence, the challenges in detecting GBC in this setting prompt the need for performing a complete and detailed histological examination of the cholecystectomy specimen, regardless of gallbladder gross appearance. Mapping of the gallbladder (GB) specimen is crucial to avoid missing cancer and to correctly diagnose the T category (Figure 1B). Likewise, to identify the N category, and predictor factor as perineural invasion (Figure 1C,D) [31,32].

Once GBC is detected on pathologic analysis, it is of vital importance to use the information available from the index cholecystectomy in the decision-making process regarding future steps in the management and the extent of oncologic extended resection (OER) for surgical candidates (Figure 2) [33]. Frequently sampled at index cholecystectomy, lymph node station 12c (Calot’s lymph node) is an independent prognostic factor for metastasis in the hepatoduodenal lymph nodes (N1 station) with no correlation to the status of lymph node stations 16, 13, and 8 (N2 station). However, Vega et al. found similar survival to N0 patients in those patients with a positive cystic lymph node (12C) only after complete lymphadenectomy and without further positive lymph nodes at OER [32].

The cystic duct margin is another important element of the pathology report of the index cholecystectomy and should be routinely reported histologically following index cholecystectomy (Figure 2). Pawlik et al. reported that compared to patients with no residual disease, patients with positive cystic duct margin were significantly more likely to have residual/additional cancer in the common bile duct (42.1% vs. 4.3%) [35]. Indeed, a higher need for resection of the common bile duct, major liver resection, total morbidity, Clavien-Dindo grade ≥ IIIa complications, and locoregional recurrence (all of which contribute to worse survival) are all strongly correlated with a positive cystic duct margin at index cholecystectomy [36].

The operative report from the index cholecystectomy should be carefully reviewed to identify risk factors in the decision making of the index cholecystectomy. Likewise, the importance of collaboration and communication amongst the referring surgeon and hepatobiliary surgeon to improve outcomes for patients undergoing GBC surgery cannot be ignored. (Figure 2). Based on registry data, intraoperative perforation of the gallbladder in patients with GBC bears a higher risk of local recurrence, peritoneal carcinomatosis, and poor prognosis [37,38,39]. By disrupting the natural barriers between the tumor and lymphatic network in the liver bed or serosa, perforation of the gallbladder at the index cholecystectomy (IC) may facilitate tumor progression [40]. GBC progression is also associated with bile spillage during routine cholecystectomy, which occurs quite frequently (25–36%) [41,42]. In challenging and more technically demanding cases involving acute or chronic cholecystitis, the perforation rate may be higher.

## 4. Non-Incidental GBC

Several investigators have reported that IGBC is associated with improved survival compared to non-IGBC [35,43,44,45] and that prior non-oncologic surgery for GBC (index cholecystectomy, IC) does not affect survival [43,44,46]. However, contrary to these previous reports, we recently reported that non-oncologic surgery during the IC can have a negative survival impact, specifically for T2b (hepatic side) tumors [14]. In some cases, there may be suspicion of GBC on preoperative imaging, but no confirmatory tissue diagnosis before exploration. These cases should be referred to a center with hepatobiliary surgical expertise. In such cases, intraoperative ultrasonography can help to accurately estimate the depth of wall invasion at the time of exploration. Moreover, in some instances, intraoperative core needle biopsy with frozen-section analysis can help solidify the diagnosis prior to committing to oncologic extended resection (OER) [47]. However, although frozen section analysis can make the intraoperative diagnosis of gallbladder cancer, the sensitivity of the frozen section has been reported as low as 64% [48,49]. In general, in the case of diagnostic uncertainty regarding GBC in centers where OER expertise is lacking, abortion of the cholecystectomy and referral to a specialized center may be prudent.

## 5. Oncologic Extended Resection

Oncologic extended resection (OER) is the recommended treatment for patients with T1b or more advanced GBC, without evidence of disseminated disease [50,51]; this entails resection of the gallbladder fossa and regional lymph node stations 12 (A, B, C, P), 8 (A, P), and 13, sampling distant lymph node station 16b1, and resection of an involved adjacent organ and common bile duct in selected cases. (Figure 3). Following OER, the reported incidence of residual cancer (RC) in patients previously diagnosed with IGBC ranges from 38.7% to 61% [15,16], with the gallbladder fossa and lymph nodes constituting the most common locations of disease [16,35,52,53,54].

OER can be performed using either the traditional open approach or, more recently, a minimally invasive approach [31,32,55,56]. Our group and others have demonstrated the safety of laparoscopic OER [55,56]; however, laparoscopic OER is performed infrequently [57,58]. To promote a wider and safe diffusion of open and laparoscopic OER, we have standardized the open and laparoscopic approach in four steps.

### 5.1. Laparoscopic Exploration and Intraoperative Sampling of Aortocaval Lymph Nodes

With the patient in the supine position for an open approach or in the French position (Figure 4) for a laparoscopic approach, start by mobilizing the hepatic flexure of the colon, caudally. The omentum adherent to the gallbladder fossa should be always resected with the liver en bloc, to avoid leaving residual cancer. Then attention was turned to the duodenum. An extensive Kocher maneuver is performed. To help with the mobilization of the duodenum, the hepatic flexure should be taken down. Next, the peritoneum is incised over the lateral border of the duodenum, freeing up the duodenum of the inferior vena cava and aorta. The wide Kocherization was achieved until both the aorta and vena cava are in the field of vision. Finally, completely dissect lymph node station 16b1 (below the left renal vein). In obese patients, it can be easier to identify the aortocaval space by coming down on the infrarenal aorta and dissecting from the aorta towards the IVC, working superiorly to the left renal vein. This lymph node station should be sent for frozen section analysis. Positive station 16 lymph nodes portend a negative prognosis and a risk–benefit reevaluation for continuing the OER at this step should be considered.

### 5.2. Regional Lymphadenectomy, including Removal of the Hepatoduodenal Ligament, Hepatic Artery, and Retropancreatic Lymph Nodes

Following the Kocher maneuver, the retropancreatic lymph nodes (station 13) should be removed. Continue the dissection cranially until the choledochoduodenal nodes (12B and P) are encountered, which will be visible on the right side of the junction of the duodenum and bile duct.

Then, the gastrohepatic ligament is incised in an avascular plane. This is continued laterally until the hepatoduodenal ligament is encountered. The common and proper hepatic artery are identified, and lymph nodes of the proper hepatic artery (8A and P) are dissected off. The common hepatic artery is dissected until the bifurcation of the left and right hepatic artery in conjunction with the anterior wall of the common bile duct, station 12A and B. The dissection of station 8A and B may necessitate controlling the left gastric vein as well as the small arterial branches near the pancreas. One should dissect all portal lymphatic tissue, but devascularization of the bile duct should be avoided.

### 5.3. Resection of the Cystic Duct in Patients with Incidental Gallbladder Cancer

Although commonly overlooked, re-resecting the cystic duct remnant is critical as it may change the operative plan (i.e., a positive cystic duct margin may dictate resection of the bile duct) and the prognosis [36]. While important, accurate identification of the cystic duct can prove to be very challenging and may lead to common bile duct injury. Avoiding the use of energy devices and using scissors instead can minimize the risk of a thermal injury to the common bile duct. This is especially true as metal clips from the initial cholecystectomy may transmit heat to the common bile duct. After the re-resection of the cystic duct stump, the specimen is sent for a frozen section. It is still a source of debate whether resection of the common bile duct in patients with a positive cystic duct margin improves survival and benefit for resection [47]. We and others have published that resection of the common bile duct failed to improve overall survival even in the subgroups of patients with positive cystic duct stump (with and without RC). However, it was associated with severe morbidity (Clavien-Dindo grade ≥ IIIa) [36]. Therefore, it is our recommendation that the decision of common bile duct resection should be analyzed case by case taking into account performance status and may be limited to select early-stage patients with a positive cystic duct margin.

### 5.4. Limited Resection of the Liver Bed or Anatomical Resection of Liver Segments IVb & V

All surgical candidates with stage 1B (The American Joint Committee on Cancer (AJCC) staging manual 8th edition) discovered after IC should have a hepatic wedge resection of the gall bladder fossa. There are no clear recommendations for the extent of hepatic resection. Some groups suggest just resecting the gallbladder fossa with others recommending a formal segment 4b/5 resection. There is good consensus, however, that patients with IGBC require at least a 2 cm wedge resection of the gallbladder fossa [59,60]. Branches of the middle hepatic vein will be encountered when the transection is carried out behind the gallbladder and will have to be controlled.

## 6. Minimally Invasive OER: Laparoscopic and Robotic Approach

Minimally invasive surgery has become a viable alternative to open surgery for OER. Several investigators have confirmed the efficacy of a laparoscopic approach reporting that blood loss, operative time, number and positive lymph nodes, R1 resection, overall morbidity, Clavien ≥ grade III complications, 90-day mortality, and recurrence patterns were all comparable to open surgery [61]. Other reports have documented that median hospital stay is significantly shorter for laparoscopic OER compared to open OER [61,62,63,64,65,66,67,68,69,70].

In general, laparoscopic OER is preferentially be used for patients that require gallbladder bed resection and lymphadenectomy only, whereas patients who need adjacent organ resection generally receive open OER. A summary of the minimally invasive OER series can be seen in Table 1.

More recently, a robotic approach to OER has shown promise. Although more data with longer follow-up is required, groups from Asia and the US have reported that robotic-assisted resection may be safe and feasible for patients with GBC [71,74,75]. Robotic surgery may have some advantages when dissecting in narrow and deep spaces [62,72,75]. A robotic approach may have advantages for lymph node removal near the pancreas and hepatoduodenal ligament, skeletonization of the hepatoduodenal ligament, hepatic artery, and celiac axis [62].

## 7. Port Resection

Our group [60] and others [66] have shown that port-site resection is not associated with improved recurrence-free survival or overall survival. This finding is supported by recent literature for gallbladder cancer [76,77,78], suggesting that recurrence of peritoneal metastases is associated more with tumor biology than with port site resection. More specifically, port-site resection does not prevent carcinomatosis or improve OS or RFS when patients with R0 resection and similar T and N categories are compared [77]. Hence, it is not recommended to routinely resect previous port sites during re-resection for gallbladder cancer.

## 8. Prognostication after OER

Recurrence following OER occurs in ~35% patients [31,79]. Among patients who developed a recurrence, around 15–53% develop a local-only recurrence, 26–66% develop distant-only recurrence, and 18–21% develop both local and distant recurrence [79,80]. The liver is the most common site of recurrence in 29–45% of patients followed by carcinomatosis (13–37%) and lymph nodes (15–25%).

Risk factors for recurrence and survival are T3-T4 disease, lymph node metastases, R1 margin, perineural and lymphovascular invasion, major liver resection, and residual cancer found after the index cholecystectomy [31,35,52,79,81]. However, (residual cancer) RC at OER is the strongest predictor of poor survival. Indeed, we have previously reported that the presence of RC is strongly correlated with poor survival (comparable to stage IV disease, even in patients with a good prognosis with R0/N0 or T1-T2 following OER) [31], and previous studies have shown that residual cancer is found in 39% to 61% of patients undergoing OER [31,35,52,82]. In contrast, patients with IGBC, but no RC at OER, may have 5-year survival rates of 73% to 85%. The poor survival in patients with RC despite otherwise good prognostic signs (R0, N0, or T1-T2) suggests that resection alone might be an insufficient therapy for these patients, prompting consideration for the additional use of neoadjuvant and/or adjuvant chemotherapy/radiotherapy.

## 9. Role of Perioperative Cytotoxic Chemotherapy in Resectable GBC

The role of perioperative chemotherapy in gallbladder cancer is emerging. There are promising data on the efficacy of chemotherapy for gallbladder cancer [83,84,85]; however, the rarity of the disease limits the conduct of randomized trials and no trials of neoadjuvant chemotherapy specifically for IGBC have been reported. The EA2197 clinical trial (Optimal Perioperative Therapy for Incidental Gallbladder Cancer (OPT-IN, NCT04559139): A randomized phase II/III trial) conducted at hepatobiliary centers throughout the US is currently accruing to fill this gap. This trial compares the combination of neoadjuvant cisplatin and gemcitabine with surgery and adjuvant gemcitabine and cisplatin versus upfront surgery with adjuvant gemcitabine and cisplatin. Of note, the chemotherapy regimen chosen in this trial was derived from the ABC-02 trial, a large multicenter phase III study on overall survival (OS; 11.7 versus 8.1 months for gemcitabine alone, hazard ratio [HR] 0.64, 95% confidence interval CI 0.52–0.80) for advanced biliary tract cancer that included 36.3% of unresectable GBC [83].

Adjuvant therapy for biliary tract cancer including, but not limited to, GBC has been more thoroughly studied. A meta-analysis by Horgan et al. showed a nonsignificant trend towards improved overall survival for adjuvant therapy (radiotherapy, chemotherapy, or both) in comparison to surgery without adjuvant therapy for patients with biliary tract cancers, including GBC (odds ratio [OR] 0.74; 95% CI, 0.55–1.01; *p* = 0.06) [86]. In this analysis, patients with microscopically involved margins (R1 resection) (OR, 0.36; 95% CI, 0.19–0.68; *p* = 0.002) and patients with lymph node-positive disease (OR, 0.49; 95% CI, 0.30–0.80; *p* = 0.004) experienced an overall survival benefit due to adjuvant therapy. In 2017, two randomized controlled phase III clinical trials investigating the role of adjuvant chemotherapy for all types of resected biliary tract cancer were reported. The PRODIGE-12/ACCORD-18 clinical trial compared adjuvant combination chemotherapy (gemcitabine and oxaliplatin) with observation, showing no significant benefit for adjuvant chemotherapy in relapse-free survival (hazard ratio [HR], 0.88; 95% CI [0.62–1.25; *p* = 0.48) [87]. Notably, gallbladder cancer accounted for only 20% of the biliary tract cancers and a majority of patients were low risk (i.e., only 13% had microscopically positive margins and 37% had lymph node metastases). Furthermore, the BILCAP clinical trial was conducted in 44 centers across the United Kingdom and included a patient sample in which 18% of cases had muscle-invasive GBC. After adjusting for nodal disease, grade, and sex (HR, 0.71; 95% CI, 0.55–0.92; *p* < 0.01) using a sensitivity analysis in a prespecified intention-to-treat analysis, this study demonstrated a benefit from adjuvant capecitabine in terms of overall survival [88]. However, RFS differed significantly between treatment groups at 2 years (HR 0.75; 95% CI, 0.58–0.98, *p* = 0.033), but not thereafter (HR 1.48, 95% CI 0.80–2.77, *p* = 0.21), suggesting that capecitabine only defers recurrence.

A recent negative trial of adjuvant chemotherapy exclusively for GBC patients was published. This randomized clinical trial comparing adjuvant gemcitabine plus cisplatin for six cycles versus no chemotherapy failed to show improved DFS and OS [89]. However, the study did not have a previous sample size calculation and the simple randomization technique allocated more patients with positive nodes and poor tumor characteristics to the chemotherapy group; therefore, no conclusion could be drawn from this study.

## 10. Targeted Therapies for GBC

Next-generation sequencing (NGS) and tumor molecular characterization have facilitated the discovery of new target pathways in ERBB receptors, PI3K/AKT/mTOR pathway, MAPK pathway, VEGF/VEFGR pathway, and DNA damage repair. A complete review of these targeted therapies is beyond the scope of this paper and the reader is referred to the literature for further details [90,91,92,93].

Perhaps the most explored targeted therapy for GBC is HER2 blockade. Specifically, GBC patients with overexpression of HER2/neu comprised 12.8% of patients and was associated with more advanced yet better-differentiated tumors [94]. Roa et al. [94] in a seminal paper with 9 patients with GBC that received either trastuzumab, lapatinib, or pertuzumab, showed that 8/9 patients had HER2 gene amplification or protein overexpression. Four patients had a partial response, one had a complete response, and three had stable disease (SD) with HER2-targeted therapy. The median duration of response was 40 weeks. Cytotoxic chemotherapy and HER2-targeted therapy showed promise when using a Trastuzumab analog that reached a 100% disease control rate and a 50% objective response rate [95].

The result of a recent study, a multicenter, phase II basket trial (MyPathway), showed a benefit with trastuzumab plus pertuzumab in a final cohort of 39 metastatic biliary cancer patients with HER2 alteration. Nine of 39 patients (~23%) had an objective response rate [96].

Currently, several ongoing trials are assessing the role of HER2-targeted therapies in BTC. Three of these trials (NCT03613168, NCT02992340, NCT02836847) are examining front-line treatment in combination with systemic chemotherapy, while an ongoing study in Japan (JMA-IIA00423) is investigating the efficacy of a combination of trastuzumab and deruxtecan for HER2-positive patients with unresectable or recurrent GBC. Furthermore, a phase II trial is currently evaluating trastuzumab plus chemotherapy in previously treated HER2-positive patients (NCT03185988).

## 11. Potential of Immunotherapy for Gallbladder Cancer

Researchers have been prompted to investigate novel treatments for gallbladder cancer by the suggestion of incremental benefit for perioperative chemotherapy reported in the studies above. Investigators are now turning their attention to the adaptive immune system, noting the relationship between the development of GBC and chronic inflammation (high mutation load) [97]. Indeed, the enhancement of the immune response through adoptive immunotherapy, vaccination, and checkpoint inhibition [98] may play a role in overcoming the problem of temporary clinical responses to traditional cytotoxic chemotherapy, which is associated with the development of resistance mechanisms that lead to disease progression and death in nearly all cases.

In this context, other studies suggest that immunotherapy for gallbladder cancer may be effective. Tumor-infiltrating lymphocyte (TIL) studies have found that infiltrating inflammatory cells, such as lymphocytes and macrophages, often surround biliary tract tumor cells [99,100,101]. Furthermore, the development of tumor-specific adaptive immune responses, mediated by infiltrating CD4^+^ and CD8^+^ T lymphocytes, are driven by tumor antigens. In other words, they play a crucial role in tumor-specific cellular adaptive immunity. Clinically, improved overall survival in biliary tumors can be predicted by the presence of dendritic cells, CD4^+^ T cells, CD8^+^ T cells, or plasma cells [100,101,102,103]. Further, investigators have suggested that the ratio of CD8^+^ lymphocytes to FOXP3+ lymphocytes (a regulatory subset of CD4^+^ T lymphocytes) could be used as a prognostic factor in biliary tract cancer [101,102,103]. Subpopulations of immune cells have not been studied in detail in gallbladder cancer specifically owing to the rarity of the disease [100]. Interestingly, however, Tran et al. described a case in which TILs from a patient with metastatic cholangiocarcinoma containing CD4^+^ T-helper-1 cells recognize a mutation in ERBB2-interacting protein. These CD4^+^ T helper cells induced a decrease in target lesions with a prolonged and durable response; moreover, a reinfusion of these TILs reproduced a similar response after subsequent disease progression [104].

Holcombe et al. investigated samples from biliary tract cancers (126 extrahepatic cholangiocarcinomas, 434 intrahepatic cholangiocarcinomas, 244 gallbladder cancers, and 11 not specified) and identified high PD-1 expression in 40% of samples and high PD-L1 expression in 15% [105]. The expression of PD-1, PD-L1 in the tumor and in the TIL, was later confirmed by other studies [27,106]. Thirty-seven of 89 patients (42%) had PD-L1-positive tumors (defined as ≥1% staining of cells in tumor nests or PD-L1-positive bands in stroma by IHC) in the biliary tract cancer cohort of KEYNOTE-28 (NCT02054806) [98]. While four of 24 patients (17%) treated with pembrolizumab (a highly selective humanized monoclonal antibody targeting PD-1) had a partial response, four achieved stable disease; five patients entered long-term treatment, including all four responders. Kim et al. [107] showed promising results of Nivolumab (anti PD-1) for GBC treatment. The phase II trial with 54 pre-treated BTC patients, of those 26% GBC patients, demonstrated a good tolerability and a 60% disease control rate in the overall case series with a median progression-free survival of 3.98 months (95% CI: 2.33–5.98) and a median OS of 14.22 months (95% CI: 6.64–NA). Today, immunotherapy for GBC has opened a door of hope in the treatment of GBC. More than 20 clinical trials in palliative treatment settings involving advanced GBC patients are ongoing around the world.

## 12. Surveillance after OER

As in most cancers, the risk of recurrence after OER for GBC is higher earlier after resection and decreases over time. Indeed, the risk of recurrence peaks at 8 months and more than 80% of the recurrence will occur within the first 18 months after OER [80]. For all patients with resected GBC, current National Comprehensive Cancer Network^®^ (NCCN) guidelines recommend surveillance with cross-sectional imaging every 3–6 months for 2 years, followed by imaging for 6–12 months up to 5 years; however, the risk of recurrence varies according to the stage of the cancer at OER. This means that the risk of recurrence was six times higher for patients with stage III–IV GBC compared to patients with stage I–II GBC, [80] suggesting that the follow-up after OER for GBC should differ in patients with stage I–II versus stage III–IV disease. For instance, for patients with stage I–II disease, we recommend that follow-up include imaging studies every 6 months for 2 years, followed by annually for up to 5 years. For patients with stage III–IV disease, we recommend that follow-up include imaging studies and tumor markers every 3 months for 18 months, followed by every 6 months up to 3 years, and annually up to 6 years.

## 13. Conclusions

Incidental GBC is the most common presentation of resectable gallbladder cancer today. Patients treated with routine cholecystectomy prior to oncologic treatment have a worse prognosis, especially those presenting with T2b tumors. Oncologic extended resection (OER) with curative intent is the only treatment to achieve long-term survival currently. Laparoscopic OER for selected patients with IGBC is safe and oncologically effective in centers with expertise. It is unlikely that resection alone is sufficient for patients with RC or stage III–IV disease. Surveillance after GBC treatment should be tailored to the patient-specific risk of recurrence. Preliminary results suggesting an incremental benefit for perioperative chemotherapy have prompted researchers to investigate novel treatments for GBC with a recent focus on immunotherapy.

## Figures and Tables

**Figure 1 cancers-14-01413-f001:**
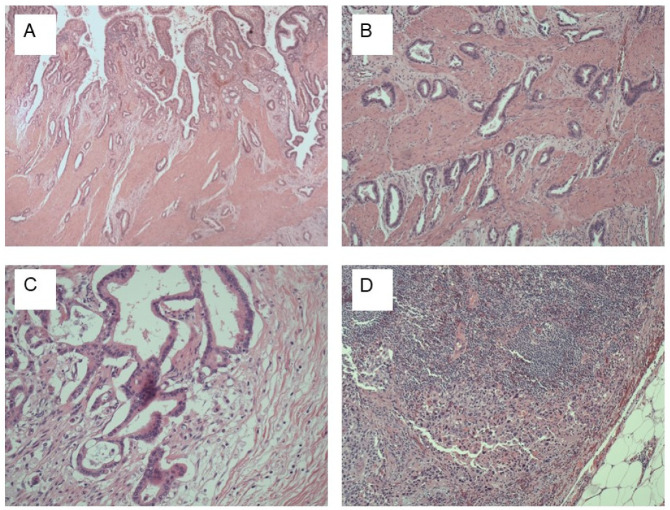
Gallbladder cancer (GBC) Histology: (**A**) GBC, biliary type, infiltrating gallbladder (GB) wall. The top of the image shows normal GB mucosa, H&E stain, ×40; (**B**) gallbladder carcinoma invading muscularis propria, H&E stain, ×100; (**C**) perineural invasion, H&E stain, ×100; (**D**): Metastatic high-grade GBC to lymph node, H&E stains, G- ×100, H- ×400.

**Figure 2 cancers-14-01413-f002:**
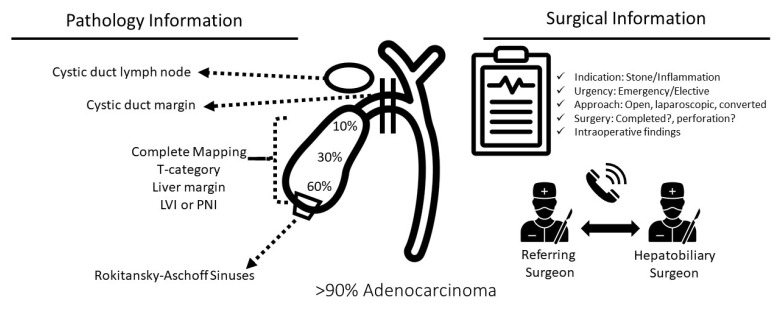
Key prognostic information in incidental gallbladder cancer (IGBC) available after the index cholecystectomy. A study of 10-year period review found that from 435 GBC patients, 391 (90%) had adenocarcinomas. Additional histologies were squamous/adenosarcoma (1.6%) and unspecified (1.1%) [34].

**Figure 3 cancers-14-01413-f003:**
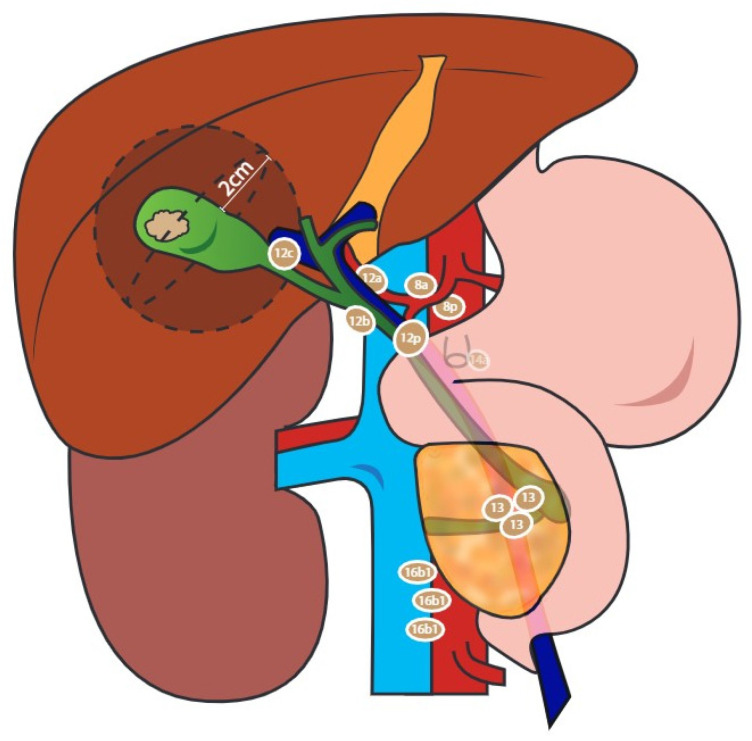
Oncologic extended resection (OER) should include sampling of aortocaval lymph nodes, specifically station 16b1; dissection of the hepatoduodenal ligament (station 12), common hepatic artery (station 8), and retropancreatic lymph nodes (station 13); limited resection of the liver bed or anatomic resection of liver segments IVb and V or rarely major liver resection.

**Figure 4 cancers-14-01413-f004:**
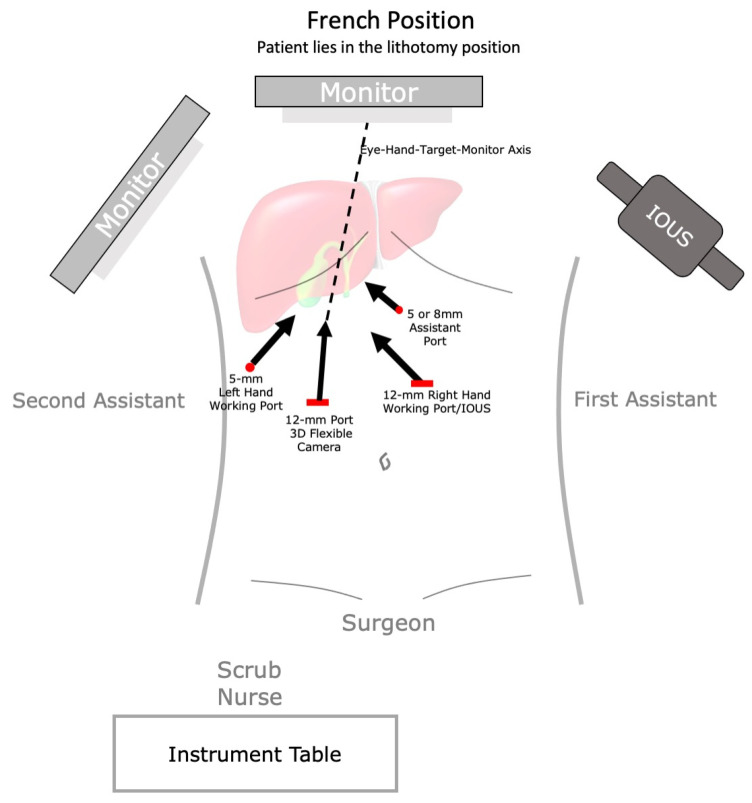
Operating room set-up and scheme of port positioning in laparoscopic OER. IOUS, intraoperative ultrasound.

**Table 1 cancers-14-01413-t001:** Minimally invasive oncologic extended resection series to date with control group.

Factors	Shen et al. [62]	Itano et al. [70]	Agarwal et al. [66]	Byun et al. [71]	Goel et al. [72]	Vega et al. [61]	Tschuor et al. [73]
Methodology							
Study period	2010–2011	2007–2013	2011–2013	2018–2020	2015–2018	2000–2017	2013–2019
Control group, open surgery	Yes, 18 patients	Yes	Yes	Yes	Yes	Yes	Yes
Patient number	5, robotic	16	24	16	27	65	20
Inclusion of non-IGBC	Yes	Yes	Yes	0	Yes	Yes	Yes
Inclusion of T3 stage	Yes	No	Yes	Yes	Yes	Yes	Yes
Surgical Outcomes							
Conversion to open	0	0	NA	0	4	13	0
Lymph node yield, median (range)	9 (3–11)	12.6 SD-3.1	10 (4–31)	3	10 (2–21)	6 (0–19)	5
Aortocaval sampling	No	No	Yes	No	Yes	Yes	No
Operative time, min median(range)	200 (120–300)	368 SD 73	270 (180–340)	198.3 (120–262)	295 (200–710)	240 (120–275)	193 (112–447)
Any morbidity	0	1	3 (13)	0	1 (7)		NA
Severe morbidity, Clavien ≥ 3	0	0	NR	0	7		2
Length of stay, days	7 (7–8)	9.1	5 (3–16)	7	4 (2–12)	4(2–18)	2.5 (0–6)
30-days mortality		0	0	NA	NA	0	
Oncologic Outcomes							
Surgical margin positive	NR	NR	0		3	3	4
Recurrence	1 (20)	0	1 (4)			11	
Overall survival	NA	NA	NA	NA	NA	3-year 87%; 5-year: 74%	OS 1-year 70.6% and 2-year 60.5%

Abbreviation: IGBC, incidental gallbladder cancer; NA, not available; SD, standard deviation.

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
