# Peer review of "Treatment of Resectable Gallbladder Cancer"

_cancers, 2022, doi:10.3390/cancers14061413_

Round 1

Reviewer 1 Report

The review article by Vega et al. comprehensively describes surgical intervention of GBC. However, the molecular alterations, systemic therapy and clinical studies are only briefly described. Given the title of the review “Treatment of Gallbladder Cancer” these topics should be extended.

  1. Although the authors write in the simple summary that “important progress has been made in the understanding of the molecular pathogenesis and systemic treatment options” the review included very little information on the molecular pathogenesis and ongoing clinical trials. Especially in the field of molecular studies there has been recently several seminal works which are worth mentioning. These include but are not limited to: Nepal et al. J Hepatol. 2021 May;74(5):1132-1144; Brägelmann et al. Hepatology. 2021 Jun;73(6):2293-2310; Pandey et al. Nat Commun. 2020 Aug 24;11(1):422; Lin et al. Nat Commun. 2021 Aug 6;12(1):4753.
  2. Several studies examined potentially druggable targets in GBC, i.e. Abdel-Wahab et al. Sci Rep. 2020; 10: 22087; Albrecht et al. Cancers (Basel). 2021 Apr 2;13(7):1682.
  3. It is not correct that IDH mutations are commonly found in GBC. Borger et al. (Oncologist. 2012;17(1):72-9) showed that IDH mutations occur in intrahepatic CCA but not in extrahepatic CCA or GBC. Similar data were also obtained by Wardell et al. (J. Hepatol. 2018, 68, 959–969).
  4. As IDH mutations are not commonly observed in GBC, treatment with Ivosidenib for IDH 1-2 mutation (line 53) should be removed.
  5. The authors are sometimes using GB carcinoma instead of GBC. It would be better for easy readability to always use GBC.
  6. As there is point 3.1 there should be also 3.2 and same holds true for 5.1. If there is no second point the heading of sections 3 and 5 may be changed.

Reviewer 2 Report

The authors reviewed the current treatment of gallbladder cancer. Here is my suggestion:
The authors introduced some mechanisms of pathogenesis of GBC at DNA level, including mutations or amplifications in key genes. At present, there have been many clinical studies on targeted therapy related with these frequently altered genes (HER2 amplification et al.) in GBC. The authors can further summarize and discuss these research progress in order to make the article more complete.

Author Response

Please see attachements

Reviewer 3 Report

This review article included several controversial issues and could be criticized.

  1. This review focused on the IGBC, not GBC in general. Therefore, title should be changed and keyword should include IGBC.
  2. In abstract, it was said that they describe a systemic approach to Indication of surgery. But, I cannot find indication of surgery in the main body.
  3. In paragraph of incidence, it was said that today, most cases of GBC are diagnosed as incidental gallbladder cancer, following elective or emergent cholecystectomy. Is there a reference for this? I cannot agree with it.  
  4. It is known that there are two pathogenesis of gallbladder cancer carcinogenesis. One is inflammatory pathway and the other is adenoma-carcinoma pathway. But you didn’t describe adenoma-carcinoma pathway. Although adenoma-carcinoma pathway is thought to be minor pathway, it should be stated because this is review paper.
  5. What do you want to demonstrate in figure 1? I think that the sentence in the manuscript and the figure are not correlate well.
  6. In figure 2, some prognostic information was not described in the manuscript. For example, the figure of referring surgeons is not described in the manuscript and >90% adenocarcinoma is not explained in figure legend.
  7. Do you think 16b1 is regional lymph node? This is somewhat controversial. Some surgeons regard 16b1 lymph node as distant metastasis and does not dissect 16b1 lymph node routinely. And if you regard 16b1 is regional lymph node, when 16b1 is positive, do you continue OER?   
  8. In 149th line, the subtitle was in selected patients, intraoperative frozen section analysis of aortocaval lymph nodes. However, selection criteria is not described in the text.
  9. If 16 lymph node is positive for malignant cells, when do you continue the OER?
  10. In 191st line, you said that it is still a source of debate whether the common bile duct should be resected in patients with a positive cystic duct margin. Then if cystic duct margin is positive, which operation do you do? Without resection of CBD, how can you do curative intended operation? Or regarding as advanced stage, don’t you continue OER?
  11. You should check the reference. For example, in 297th line, reference number is 93. But reference number 94 is right for this sentence.              

Round 2

Reviewer 1 Report

The authors have addressed all of my previous points to my full satisfaction. However, there are some minor grammatical and typing problems in the revised version.

Line 55: revealed instead of reveals

Line 63: TMB-H was not introduced

Line 225: into instead of in to